# Characteristics of Train–Pedestrian Collisions in Southwest China, 2011–2020

**DOI:** 10.3390/ijerph19106104

**Published:** 2022-05-17

**Authors:** Zizheng Guo, Zhenqi Chen, Jingyu Zhang, Qiaofeng Guo, Chuanning He, Yongliang Zhao

**Affiliations:** 1School of Transportation and Logistics, Southwest Jiaotong University, Chengdu 610031, China; guozizheng@swjtu.edu.cn (Z.G.); chenzhenqi@my.swjtu.edu.cn (Z.C.); guoqiaofeng860516@163.com (Q.G.); 2National Engineering Laboratory of Integrated Transportation Big Data Application Technology, Chengdu 611756, China; 3National United Engineering Laboratory of Integrated and Intelligent Transportation, Southwest Jiaotong University, Chengdu 611756, China; 4Comprehensive Transportation Key Laboratory of Sichuan Province, Chengdu 610031, China; 5CAS Key Laboratory of Behavioral Science, Institute of Psychology, Beijing 100101, China; 6Department of Psychology, University of Chinese Academy of Sciences, Beijing 100049, China; 7Chengdubei Railway Station, China Railway Chengdu Group Co., Ltd., Chengdu 610512, China; hchuanning@163.com; 8Traffic Control Center, China State Railway Group Co., Ltd., Beijing 100844, China; zhaoyongliang0328@163.com

**Keywords:** railway accident, train–pedestrian collision, descriptive analysis, accident records

## Abstract

Although train–pedestrian collisions are the primary source of railway casualties, the characteristics of this phenomenon have not been fully investigated in China. This study examined such collisions in the Greater Sichuan-Chongqing area of China by conducting a thorough descriptive analysis of 2090 incident records from 2011 to 2020. The results showed that such collisions have declined gradually over the past decade, but the fatality rate remains high. We found that such collisions were more likely to happen to men, senior citizens and people crossing the tracks and that they occurred more frequently in the morning. While collision rates dropped in February, collisions were more likely to occur in December. In contrast to the situation in Western countries, weekends were not related to increased occurrence. The absence of a protective fence led to a higher collision rate, but level crossings are no longer a concern since most such structures in China have been rebuilt as overpasses. Mild slopes and extreme curvatures were also found to increase the occurrence of such collisions. Freight trains were most likely to be involved in train–pedestrian collisions, and collisions caused by high-speed trains were rare both absolutely and relatively. However, when collisions did occur, higher train speeds were linked with higher fatality rates. The findings suggest that patterns of train–pedestrian collisions in China differ from those in the Western world. This difference might be caused by differences in culture, geography, weather and railway development policies. Future research directions and possible preventive measures are also discussed.

## 1. Introduction

Safety is essential for railway development, and accident prevention is the primary concern in railway operations. Although large-scale railway accidents, such as collisions between trains or derailments of trains, easily attract the attention of the media and the public, the actual occurrence of such incidents is rare. According to European Union statistics, train–train collisions and derailments account for only 5% of all railway accidents [1]. In contrast, accidents caused by pedestrian intrusion onto the tracks constitute a significant percentage of railway accidents and should warrant more attention. Previous studies conducted mainly in Western countries have shown that train–pedestrian collisions have become the leading cause of casualties in railway accidents [2,3,4,5]. For instance, in the United Kingdom, the number of deaths from illegal intrusions of pedestrians onto railways is three times larger than other types of railway-related deaths [6]. In Sweden, approximately 80–100 people are killed each year in train–pedestrian collisions [7], and in the United States, the number is about 500 [8].

The severity and impact of train–pedestrian collisions are generally greater than those of road collisions. Due to their high speed, long braking distance and inability to change directions, trains are more likely to cause severe injuries and deaths than vehicles on the road. In the United States, the casualty ratio was 54.1% in 2021 [8], and in Canada, this figure was as high as 67.2% [9]. In addition, since the scene of train–pedestrian collisions may be very dreadful, people involved in such accidents, such as train drivers, passengers, bystanders and rescuers, might experience particular psychological trauma from which they find it difficult to recover [10,11,12,13,14].

Due to the dreadful consequences and the prevalence of pedestrian intrusions, it is necessary to identify the causes of these collisions so that administrative personnel can take appropriate preventive measures. To date, most studies on this issue have used accident record data and descriptive analysis to reveal certain influential factors. These factors can be categorized in four groups: the characteristics of pedestrians, the time of the collision, the structure of the track and the type of train. Because the demographic information of the victims and the time of the collisions are always included in surveys, these two factors are most commonly analyzed.

Among the characteristics of the pedestrians in these cases, one consistent finding across different studies has been that men are more likely than women to be involved in such collisions [15,16,17,18]. In terms of age, most studies have found that young adults are more likely to be involved than other age groups. For example, Savage’s studied 471 train–pedestrian collisions in the United States in 2005 and pointed out that people aged 16–45 constituted the majority of such collisions [17]. Mohanty and Patnaik’s research on 88 collisions in India showed that pedestrians between 21 and 40 years old were the most common victims of train–pedestrian collisions [19]. In Finland, the most vulnerable age group was even younger at 10–29 years old [18].

Another important pedestrian characteristic is their intention. While many pedestrians involved in these collisions were mere trespassers, others may have intentionally used the railway as a means to commit suicide. In some countries, the rate of suicide is very high. For instance, in Sweden, researchers can distinguish whether an incident was a suicide on the basis of a police report [16]. In another study, researchers analyzed train suicides on an international level and found that train suicides accounted for 1–12% of all suicides [20]. Although there are established criteria for judging railway suicides, it is still difficult to distinguish a suicide attempt from other types of intentions because most collision records provide insufficient information [18]. Precrash behaviors, if accurately documented, can help identify an individual’s purpose. For example, lying on the track is typically considered a pre-suicide behavior, while jumping off the track is not. Most studies that have used this type of data have found that walking, sitting or lying on the track were the most frequent behaviors in cases of collisions [21]. Even though these behaviors are more dangerous than others, it is still difficult to determine whether they can be regarded as precise proxies of suicide. As the records are based mainly on the train drivers’ narratives, it is difficult to verify whether “lying on the track” describes a scene accurately. Even if it is correct, understanding whether an individual instance is a suicide attempt or an involuntary fall is not an easy task.

Additionally, few studies have examined other pedestrian features. In two studies of situations in the United States, most train–pedestrian collisions were found to be related to alcohol or drug usage [22,23]. In another study, Pelletier found that most victims were area residents rather than homeless people or temporary residents, suggesting that loitering may not have been the cause of the collisions [5].

When analyzing the times of such occurrences, current findings can be organized into the time of year, day of the week or date. In terms of the time of year, some studies have found that such collisions are more likely to occur in spring and summer, possibly as a result of increased outdoor pedestrian activities [16,24]. However, a study in Finland found no difference among the twelve months [18]. For the day of the week, studies have found that collisions are more likely to occur in the second half of the week (Thursday to Sunday) or on weekends, showing the influence of typical working habits in industrialized countries. However, collisions in New Zealand are evenly distributed [25].

Some features associated with the locations of collisions also played an important role. Interestingly, many researchers have focused on collisions at railway level crossings because level crossings are where people can legally cross the railway and are frequently the sites of accidents. Therefore, many studies have focused on analyzing the frequency and severity of collisions at level crossings [26,27] and proposed reasonable improvement measures [28].

Table 1 is a partial summary of the references cited in this article. Although there are several existing works on pedestrian–train collision accidents and their causes, many issues have not been fully studied and resolved.

First, most of the published papers focus only on situations in Western developed countries. However, railroads also play an important role in developing countries, where train–pedestrian conflicts can be more severe because of denser populations and fewer protective facilities. Among developing countries, China is worth investigating for several reasons. First, as a major railroad country, China has the second-longest railway mileage in the world with 146,000 km and is still expanding its networks. The operational speed of the trains has become increasingly faster. Second, as one of the most populous countries in the world, China has experienced dramatic industrialization and urbanization in recent decades. As a result, there are serious conflicts in the use of land. Previously uninhabited areas around railways have been developed due to economic growth; thus, new railways must cut across certain habitual local routes, such as people’s living areas and farms. All these issues, which may differ from those examined in Western societies, may increase the likelihood of train–pedestrian conflicts. Skládaná et al.’s research on the Czech Republic explored a similar situation in Eastern Europe [29].

Third, although many studies have investigated risk-contributing factors in the personal and temporal domains, how the characteristics of places and trains can influence train–pedestrian collisions has not been fully explored. Some protective facilities (fences) have been important in preventing collisions in small-scale experimental studies, but no research has investigated their effects based on a large set of data. While certain places, such as level crossings, are considered very dangerous, factors that might influence collisions on the track have not been fully studied, probably due to the difficulty in obtaining relevant data. However, evidence accumulated from road collision research has shown that road characteristics such as slope and curvature can influence the control of trains or detection of potential trespassers [30,31,32]. Although previous studies have presented no direct evidence yet, we believe these properties might have certain effects.

Finally, some papers have compared the observed distribution of collisions with the baseline distribution in interpreting the data. For example, when concluding whether men are more likely to become involved in a collision than women, we should compare the observed distribution in the collision report with that in the regional population. This comparison is particularly important for studying situations in a society such as China because it has witnessed major changes in demographics.

To fill these research gaps, we examine train–pedestrian collisions in China for the first time. Our analysis was based on a 10-year collection period of collisions in Southwest China, a crowded, mountainous and inland region with ethnic, geographic and economic diversity. To depict a full picture of these records, we also obtained other data types from various sources. We compared the demographic data in the acquired forms with those of the general social survey to better understand collision characteristics in China. In addition, we collected structural information about the tracks from maintenance departments in order to explore the influence of track characteristics, including the slope, curvature and presence of fences.

## 2. Method

### 2.1. Data Sources

The collision record data used in this article came from China Railway Chengdu Group Co., Ltd (Chengdu, China). The data were collected by the Chengdu Railway Bureau Dispatching Office, which recorded collisions between trains and pedestrians from 2011 to 2020 within the jurisdiction of the Chengdu Railway Bureau in the greater Sichuan-Chongqing area of China. For technical reasons, some of the casualty data for 2018 (January to June) were missing. The jurisdiction area includes the entire territory of Sichuan Province and Chongqing City and parts of Guizhou, Shaanxi and Yunnan Provinces, as shown in Figure 1. The region of interest has a total area of 740,500 square kilometers, with a population of approximately 155,891,700 and a railway mileage of 17,893.88 km. The database included 2090 independent collisions causing 2139 casualties; the injury or death status of three of these incidents was unknown. In most cases, information about the time, location and severity of the collision and the sex and age of the victims was recorded. Some entries contained additional information, such as train type, train speed before the collision, the presence of fences and victims’ precrash behaviors.

We were authorized to utilize the track structure database that includes all relevant information about the rail tracks in this region, e.g., the slope, curvature and distance to the nearest bridges, tunnels or stations based on the collision location in the records.

### 2.2. Coding

Since the original data were all text-based records, we performed data transcription and coding. Information of interest was divided into four categories: pedestrian characteristics, time features, train characteristics and track characteristics. Table 2 summarizes all the variable information in these four categories, part of the data missing was due to simplified recording requirement in the early years (e.g., speed, pre-accident behavior, etc.).

**Table 2 ijerph-19-06104-t002:** Variable information divided into four categories.

Classification	Variable	Definition of Variables	Available in N Records
Pedestrian characteristics	Age	Age of casualties in the collision	2054
	Sex	Sex of casualties in the collision	2122
	Precrash behavior	Behavior or posture of casualties before the collision (for details, see Table 3)	438
Time characteristics	Year	Year when the collision occurred	2139
	Month	Month in a year	2139
	Workday	Day of the week	2138
	Time of day	Hour of the day (24 h)	2139
Train characteristics	Train type	Type of train (operating trains (such as crane or maintenance trains), high-speed passenger trains, regular-speed passenger trains and freight trains)	2118
	Speed	Train speed at time of collision	453
Track characteristics	Slope	Slope of the track where the collision occurred (schematic diagram is shown in Figure 2)	1815
	Curve radius (R)	Curvature of the track (schematic diagram is shown in Figure 3)	1667
	Location	In a station or along a line	2139
	Fence	Whether protective equipment was present at the site	771
	Level crossing	Whether the collision occurred at a level crossing	2139

### 2.3. Data Analysis Protocol

We first analyzed how collision frequencies were distributed along with each categorizing variable. Second, we analyzed the interaction between variables to influence collision occurrence. In addition, we also analyzed the severity of the crash by individual variables. We used mainly descriptive statistical methods, and a chi-square test was performed to compare the real distribution with the expected distribution when necessary. All analyses were performed in SPSS 23.0.

## 3. Results

### 3.1. Collision Frequency across the Levels of Each Variable

We first analyzed how collision frequencies were distributed along with each categorizing variable. Table 4 summarizes the distributions. Then, we used chi-square analysis to examine whether the distribution differed from our expectations. For all variables, we compared the actual distribution with an equal distribution hypothesis. For some variables, if we were able to obtain their baseline distribution (e.g., sex, slope, curvature), we also compared it to the baseline distribution.

#### 3.1.1. Age

The number of casualties across each age group differed significantly from an even distribution (χ^2^(6) = 123.634, *p* < 0.001). Older people (over 70) accounted for the largest proportion (19.70%). As people over 65 in the Greater Sichuan-Chongqing area of China represent only 15.60% of the total population [33], the observed percentage of elderly people suggests an elevated risk.

#### 3.1.2. Sex

The number of casualties across the two sexes differed significantly from an even distribution (χ^2^(1) = 62.287, *p* < 0.001). Men accounted for the largest proportion (62.4%). As men represent only 51.24% of the total population, the observed percentage of men suggests an elevated risk (χ^2^(1) = 46.813, *p* < 0.001).

#### 3.1.3. Precrash Behaviors

Due to missing data, only 438 records were used to analyze the behaviors of pedestrians before the collision. The number of casualties differed when the pedestrians showed different precrash behaviors (χ^2^(6) = 347.087, *p* < 0.001). The most common behavior before the collision was a dynamic one, walking or running across the track (39.5%), followed by another dynamic behavior, walking along the track (25.6%). Static behaviors, such as standing, squatting or lying on the tracks, occurred at similar rates (approximately 11.00%).

#### 3.1.4. Year

The number of casualties showed a significant downward trend in the last decade (χ^2^(9) = 227.026, *p* < 0.001). In particular, this number had declined rapidly since 2015. Nevertheless, note that the official statistics for 2018 were only for half of the year.

#### 3.1.5. Month

The distribution of collisions was not even across the twelve months (χ^2^(11) = 23.912, *p* = 0.013). The collision rate was lowest in February (6.20%) and peaked in December (10.00%).

#### 3.1.6. Workday

The collisions were distributed evenly across the seven days of the week (χ^2^(6) = 6.573, *p* = 0.362). No evidence suggested that collisions are more likely to happen during weekends.

#### 3.1.7. Time of Day

The collisions were distributed unevenly across the day (χ^2^(23) = 251.362, *p* < 0.001). The occurrence of collisions was concentrated in the daytime from 7:00 a.m. to 7:00 p.m. (64.10%). It reached the highest rate from 10:00 to 11:00 a.m. (6.60%) and the lowest rate from 3:00 to 4:00 a.m. (1.70%).

#### 3.1.8. Train Type

Collisions occurred at different rates across different types of train (χ^2^(3) = 2123.092, *p* < 0.001). Freight trains (60.20%) and regular-speed passenger trains (37.10%) account for most of the collisions (97.30%), but collisions involving high-speed passenger trains are very rare (0.50%). Since the proportion of the freight trains, the regular-speed passenger trains and the high-speed passenger trains in the total running trains were approximately 66%, 16% and 18%, respectively (to note, this percentage was deduced from multiple different sources), it seems that the regular-speed passenger trains showed an elevated risk, but the risk of HSP train is much lower than expected.

#### 3.1.9. Speed

Due to missing data, only 453 records were available for this analysis. The frequency of crashes in different speed zones is different (χ^2^(4) = 29.815, *p* < 0.001). At the time of the collisions, most trains were running at a speed between 61 and 70 km/h (28.00%), with the lowest number of trains above 80 km/h (12.80%).

#### 3.1.10. Slope

Collisions occurred at different rates across different slopes (χ^2^(6) = 30.204, *p* < 0.001). Most collisions happened on slopes (either uphill or downhill) (88.40%). The collision rate peaked (15.9%) on steep uphill slopes (slope > 6‰). The baseline of the slope and the proportion of different slopes in the collisions are shown in Table 5. By comparison with the baseline, the difference in the proportion of collisions occurring on different slopes was also significant (χ^2^(6) = 125.796, *p* < 0.001). Figure 3 shows that the frequency of collisions was lower than expected for very steep uphill and downhill sections (slope > 6‰ and slope < −6‰). In contrast, in the gentle slope sections (−6‰ ≤ slope < 0 and 0 < slope ≤ 6‰), the probability of a collision was higher than the proportion of those sections to the mileage of the whole line. The largest discrepancy between the collision rate and the mileage ratio was achieved in the slightest slopes (−3‰ ≤ slope < 0 and 0 < slope ≤ 3‰).

#### 3.1.11. Curve Radius

Collisions occurred at different rates across different curve radii (χ^2^(6) = 2377.761, *p* < 0.001) and were more concentrated in straight sections (51.10%). The baseline of the curve radii and the proportion of different curve radii in the collisions are shown in Table 6. By comparison with the baseline, the difference in the proportion of collisions occurring on different curve radii is also significant (χ^2^(6) = 1181.385, *p* < 0.001). Figure 4 shows that the probability of collisions on straight lines and sections with large curve radii is lower than expected. When the curve radius is less than or equal to 1000 m, the probability of collisions is much higher than the proportion of that curve radius to the entire line. Especially when the radius is less than 500 m, the probability of a collision is three times that of the whole mileage. We concluded that a smaller curve radius and a sharper bend can lead to more dangerous results.

#### 3.1.12. Fence

Due to missing data, only 771 records were available for this analysis. Whether there is a protective fence at the location of the collision differs significantly and is unevenly distributed (χ^2^(1) = 493.760, *p* < 0.001). The vast majority of collisions occurred without a fence (90.00%), and only 10.00% closed the track.

#### 3.1.13. Location

The number of collisions in different locations varied significantly (χ^2^(1) = 1314.787, *p* < 0.001). Most of the collisions occurred along the line between stations (89.20%), whereas collisions occurring at stations accounted for only 10.80%.

#### 3.1.14. Level Crossing

Only two collision incidents in the database occurred at level crossings, accounting for 0.09% of all incidents.

### 3.2. Interaction between Variables to Influence Collision Occurrence

#### 3.2.1. Age by Sex

To analyze the demographic information of collisions more clearly, we performed a joint tabulation analysis of age and sex variables. The results showed significant differences in the sex division of different ages (χ^2^(6) = 88.841, *p* < 0.001). The bar figure clearly shows that between the ages of 18 and 50, the proportion of casualties is much higher among men than among women. The difference in this proportion gradually shrinks with age. After the age of 70, the number of female casualties surpassed the number of male casualties (see Figure 5).

#### 3.2.2. Precrash Behavior by Location

We also performed a chi-square test analysis of precrash behavior and the location of the collision (see Table 7). The results showed that precrash behavior was significantly correlated with the location of the collision (χ^2^(6) = 71.683, *p* < 0.001). Whether in the station or along the line, crossing the track was the most frequent behavior, followed by walking along the track. The main difference was that climbing over and creeping under the train happened only inside stations (see Figure 6).

### 3.3. Analysis of Collision Severity

We recorded a total of 2136 data points of casualties in collisions, among which 963 people were injured, accounting for 45.10%, and 1173 died, accounting for 54.90%. Our results show that when a train–pedestrian collision occurs on a railway, it is more likely to cause a death than an injury.

We conducted a chi-square test on several independent variables and our dependent variable, collision severity, exploring whether these characteristic variables in collisions are significantly correlated with the severity of collisions. This analysis can be helpful in determining approaches to lessen the severity of collisions.

#### 3.3.1. Pedestrian Characteristics and Severity

Pedestrian characteristics included sex, age and precrash behavior. The results showed that neither the sex nor the age of casualties was significantly associated with collision severity. As the frequency of some behaviors was less than five among the behavior variables in the station, it was impossible to perform a chi-square test for them. Nevertheless, we calculated the proportion of injuries and deaths in different locations for different behaviors to explore this question (see Table 8). The death rate for standing in the station was 100%, but it was not an object of our analysis and conclusion due to the small data quantity. The death rate for lying in the station and along the line was very different, but the death rate for other precrash behaviors, such as crossing the line and squatting on the tracks, was relatively similar in different locations (see Figure 7 and Figure 8).

#### 3.3.2. Time Characteristics and Severity

The time characteristics that we considered were the year, month, day of the week and time of day. Among them, only the year had a significant correlation with the severity of the collision (χ^2^(9) = 55.686, *p* < 0.001), and the rest of the temporal variables had no statistical significance. As shown in Figure 9, the number of injuries exhibited a decreasing trend from year to year, while the change in the number of deaths was more complex, and no clear pattern emerged. Overall, the total number of casualties still showed a declining trend.

#### 3.3.3. Train Characteristics and Severity

The severity of the collisions did not differ across different train types (χ^2^(1) = 2.940, *p* = 0.092). However, the severity varied significantly across different speeds. The higher the speed was, the higher the fatality rate (χ^2^(4) = 53.329, *p* < 0.001) (see Figure 10).

#### 3.3.4. Track Characteristics and Severity

The chi-square test results showed no significant relationship between the characteristics of the track and the severity of collisions. Collision severity was not affected by changes in the radius of the slope (χ^2^(6) = 6.189, *p* = 0.402), location(χ^2^(1) = 0.554, *p* = 0.457) or curve radius (χ^2^(6) = 9.185, *p* = 0.163).

## 4. Discussion

The main purpose of this study is to describe the basic characteristics of train–pedestrian collisions in the Greater Sichuan-Chongqing area of China and provide possible suggestions for future research and practice. Using collision records from 2011 to 2020 in this area, we analyzed how collision occurrence and severity were distributed across pedestrian-, time-, train- and track-related characteristics. Several findings are worth discussing.

First, there were 2090 train–pedestrian collisions in this area over the selected decade, resulting in a total of 963 people injured and 1173 people killed. Given the large population in this area (millions), the annual collision rate per million people was 1.4. It should be noted that this rate is lower than that in other countries, such as Finland (11.2) [34], the United States (3.1) [8,35], and the European Union (6.2) [36,37]. In addition, the fatality rate was 54.9%, which is similar to that of the United States (54.10%) [8] and slightly lower than that of Canada (67.20%) [9], emphasizing the severe consequences of such collisions across different countries.

Regarding pedestrian-related characteristics, we found that men were more likely than women to be involved in train–pedestrian conflicts. This finding is consistent with previous studies in other countries [18]. A possible explanation is that men are more risk-seeking due to male hormones such as androgen [38]. Further analysis corroborated this explanation because the sex differences diminished as the victims became older, as illustrated in Figure 6. As the secretion of sex hormones decreases during the aging process, men and women become equally likely to be involved in such collisions.

When analyzing the effect of age, we found that the oldest group (61–70) was the largest group of victims. This finding differed from those of studies in Western countries, where most victims were young and middle-aged people [18,19,24]. The difference cannot be explained by age composition because although the aging population is rapidly increasing in China, the ratio of senior citizens is still lower in the Chinese population than in most Western societies [39]. One possible reason is that the rapid expansion of the railway might cause certain conflicts with the habits of local people [29]. Whereas younger people might adapt to these changes very quickly, older people are less likely to change their trip habits and thus are more likely to be involved in this kind of collision. Future research could examine whether older people in this area are less aware of new changes in the infrastructure or are more likely to cling to their old habits at higher risk. If so, it might be important to design age-friendly educational campaigns to prevent such collisions.

In terms of precrash behavior, we found that the leading behaviors before collisions were running across the tracks, followed by walking along the tracks, either on the line or in the station. This finding is consistent with previous research in Western countries [21]. Further analyses showed that the relationship between precrash behaviors and collision severity differed across locations. When the collisions occurred in the station, the fatality rate was approximately 58.70% and reached the highest level when the victims ran across the tracks (64.70%). However, when they happened on the line, the overall fatality rate was slightly higher (65.80%), and the death rate was highest when the victims were lying on the tracks (82.60%). We believe the reason is that in the station, railway personnel can prevent people lying on the tracks from being hurt. Since the lying action is generally time-consuming, there would be enough time to notify the driver to brake or to forcibly remove people from the tracks. However, when the behaviors happen along the line, there are no personnel to prevent them, which could be the reason for the difference in fatality rates for people lying on the tracks. On the other hand, if a person runs across the tracks, railway personnel are unlikely to have time to respond, whether they observe the behavior or not. This reduces the difference in the fatality rate across locations.

For time-related characteristics, we first found that the collision rate in this area was decreasing annually. One possible reason might be the increased use of fences. Although we do not have actual statistics for the yearly installation of new protective equipment in this area, there has been a continuing effort to improve the protective infrastructure. According to Chinese government documents, since 2014, railway lines with a design speed of more than 120 km per hour have been completely closed through measures such as closed facilities and warning signs [40]. Previous studies have suggested that fencing is among the most effective measures to prevent pedestrian intrusions [41], and we believe that such measures are equally effective in China. However, the death rate in collisions appears to be increasing, and we have a few guesses about this trend. First of all, with the development of technology, the speed of trains has gradually increased during this decade, which may increase the severity of collisions. However, we cannot prove this assertion using our own data directly because the speeds in the early years were not fully recorded, as mentioned in a previous response. Secondly, more protective nets were installed after 2014. While this can reduce the number of less motivated trespassers (e.g., who only want to take a short cut), it might not reduce the number of very motivated intruders. For example, some people may deliberately break the protective net to commit suicide or just show off. As these behaviors are more related to severe consequences, the increase of death rate is reasonable. A minor but also possible reason is that the driver may reduce their degree of vigilance when driving along the line with protective nets installed. They would not expect people to appear on the track and have slower reactions.

In analyzing the monthly distribution, we found that collisions did not occur uniformly throughout the year. We observed that the collision occurrence reached its lowest value in February and its highest value in December. Previous studies have suggested that low temperatures and the Chinese spring festival might reduce the time that people spend outdoors. This may explain why collisions occurred with the lowest frequency in February. However, the highest occurrence rate in December cannot be explained. A possible interpretation is a special weather condition in these mountainous areas called swarm fog, which happens mostly in December. As this kind of fog can reduce the visibility range for drivers and pedestrians, it might increase the occurrence of collisions in this month [42]. It might be fruitful for future studies to examine this possible influential factor by incorporating experts in meteorology.

We further found that collisions occurred equally across days of the week. Notably, this finding is different from the findings of studies in Western countries. While researchers from different countries will have different results, according to national rail collision data in the United States and previous research, the risk is higher on Fridays and Saturdays, and 49.2% of suicides and 65.7% of accidents in Finland occurred on weekends (Friday to Sunday) [18]. This difference can be explained by the difference in the data of the population engaged in agriculture. Office workers do not need to work on weekends and can move around at will, while agricultural workers carry out agricultural activities according to the solar and weather conditions. There are no fixed rest days within a week, so the collision data within a week will not be concentrated on a particular day.

Within a day, collisions were more likely to happen in the daytime and reached the highest rate in the morning (10:00–11:00 a.m.) and the lowest late at night (3:00–4:00 a.m.). Such a pattern seems to be very similar to the human circadian rhythm, which governs the outdoor activities of pedestrians. However, the train schedule is a confounding variable here. As more trains run in the daytime than in the night, it is difficult to draw a firm conclusion that pedestrians’ travel behaviors are the only cause of such an hourly distribution. Future studies may further examine this possibility by controlling the quantity of train flow.

Regarding train-related characteristics, train types were found to play an important role. We discovered that freight trains were involved in most of the collisions (0.20%), followed by regular-speed passenger trains (37.10%). This pattern is very different from that in Western countries, where commuter trains were most likely to cause collisions [24]. This ratio is in accordance with the proportion of freight trains to all running trains (66.90%) [43], which is associated with the manufacturing and logistic power of China. It is worth noting that although high-speed passenger trains (HSPTs) represent a substantial proportion of all running trains (10–22%) according to two different sets of statistics [44,45], the number of collisions caused by HSPTs is very low (0.50%). The reason is that high-speed rail lines are built on a highly elevated base, and protective nets have been installed throughout the entirety of the lines, making it difficult for pedestrians to intrude.

We also found that train speed can influence collision severity. Victims are less likely to survive when trains run at higher speeds. Although this finding is seemingly quite commonsensical, to the best of our knowledge, we are the first to report the actual relationship between train speed and collision severity. As we also suffer from the problems of limited data, we must be aware that any conclusions about the speed might be confined to situations of recent years; future studies may use a new apparatus to accurately record the speed before the collision and use this finding to enact new speed-limit regulations in areas where collisions happen more frequently.

Regarding track-related characteristics, when analyzing the location, we found that more collisions occurred along sections between stations, and 10% of collisions occurred within stations. However, the mileage of the line owned by stations represents far less than 10% of the mileage of the entire line. Although the absolute number of collisions occurring at stations is low, the danger level is worthy of attention. This danger may be because, despite the management and monitoring of staff in stations, pedestrians can still intrude onto the track without any hindrance. Moreover, in stations, the times that pedestrians and trains arrive at the track are almost the same. When a train is coming, the density of pedestrians is much higher in the station than along the line.

Finally, regarding level-crossing collisions, in the past ten years, only two collisions occurred at level crossings within the jurisdiction of the China Railway Chengdu Group Co., Ltd. Due to the small number of collisions, there is not enough data to analyze them individually. This is very different from the relevant conclusions in other countries. In the United States, a total of 1749 collisions occurred at road-rail level crossings in 2021 [46]. An Australian study found that collisions at level crossings cause significant economic losses of more than AUD$ 116 million per year [47]. In China, there have been almost no collisions at railway level crossings; from the documents we obtained from the China Railway Chengdu Group Co., Ltd., we found there are 48 level crossings along all the tracks in this area (17,893.88 km). Therefore, the number of level crossings per 100 km is 0.268. According to public data in the United States, the United States has a total of 203,778 level crossings (about 250,000 km) [48], and the number of level crossings per 100 km is 81.5, which is much higher than China. This can prove that measures to reduce level-crossing accident rates, such as reestablishing crossings and building underpass tunnels or pedestrian bridges, have been effective [49].

We might be the first to examine the influence of curvature and slope on train–pedestrian collisions, and we obtained some interesting findings. First, by comparing the collision distribution from the baseline (the actual proportion of different curvatures), we found that collisions were more likely to occur on sharper bends. The reason might be that a large curvature (small curve radius) may make it challenging for drivers to control the train and reduce the visibility range. Although studies in automobile driving have suggested that large curvatures may increase the workload of car drivers [28], few studies have examined their possible influence on train drivers. Future studies may use train simulators to further explore such influences. Second, in terms of track slope, we found that collisions were more likely to occur on gentle slopes, either upward or downward, than on level tracks. However, the collision rate dropped again on the steepest slopes. This cannot be explained by the fact that steeper roads cause greater psychomotor demand for drivers. One possible explanation might be that drivers may have different risk perceptions and vigilance levels when facing different slopes [50]. When the slopes are gentle, although they still cause certain problems in controlling the train, the drivers may not be aware of the risk factor. However, when the slopes are very steep, the drivers will be highly attentive and use the highest standard to ensure safety when operating on these more dangerous sections (e.g., reducing the speed of the train). Future studies may examine drivers’ vigilance level and risk perception on different slopes to test this explanation. If this is the case, it might be useful to develop certain training or warning systems to improve drivers’ safety awareness.

## 5. Limitations and Future Direction

Although our study obtained remarkable findings, some limitations must be noted as well. First, although the present research built a large data set in this area and used some new types of data (e.g., track curvature and slope), it still suffered from missing or incomplete data, as did many previous studies. Future studies may benefit from using better-designed recording forms or modern digital on-train records. Second, as ours was the first attempt to analyze train–pedestrian collisions in China, we used mainly descriptive statistical analysis to create an overall picture of this phenomenon. As we found several remarkable differences between China and Western countries, future studies may use more advanced techniques, such as Poisson regression, and consider temporal and spatial heterogeneity. Third, we investigated collisions in a very large area, but it is still only part of China. As this area is mountainous and relatively less developed, the conclusions of the study cannot be readily generalized to other regions of China or other developing countries.

## 6. Conclusions

As the first to examine train–pedestrian conflicts in China, the present study showed that train–pedestrian collisions are an important safety concern in China, but the situation is improving. We found several factors that have been identified in previous studies, such as being male, certain crossing behaviors, occurrence in the daytime, not having fences, occurrence around the winter new year holidays (February) and higher speed, to be influential. However, level crossings and occurrence on the weekend were not risk factors. We also found new risk factors, such as being elderly, occurrence in December, large track curvature and mild slopes. These findings suggest that certain cultural, geographic, weather and infrastructure-related issues should be considered to understand the causes of pedestrian–train conflicts. Educational policies and other preventive measures should be enacted based on these findings.

## Figures and Tables

**Figure 1 ijerph-19-06104-f001:**
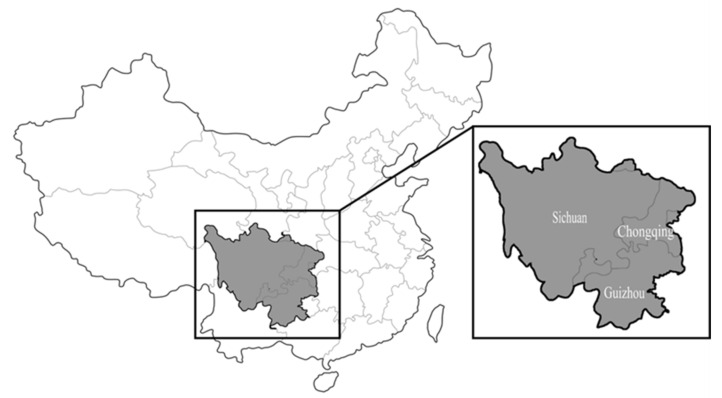
Area under the jurisdiction of Chengdu Railway Administration.

**Figure 2 ijerph-19-06104-f002:**
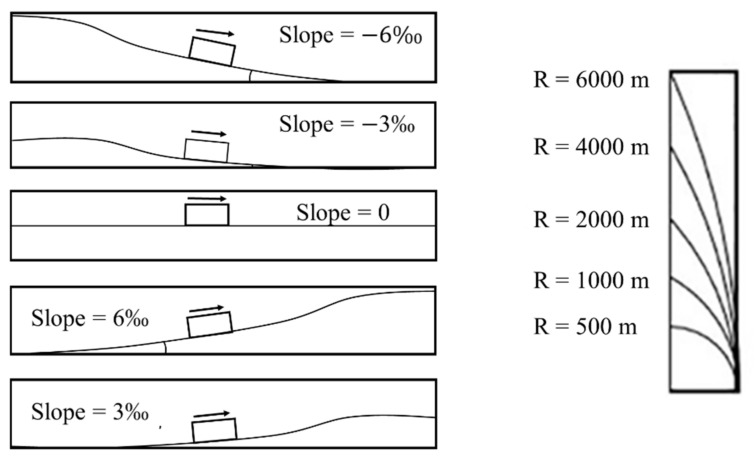
Schematic diagram of slope and curve radius (R).

**Figure 3 ijerph-19-06104-f003:**
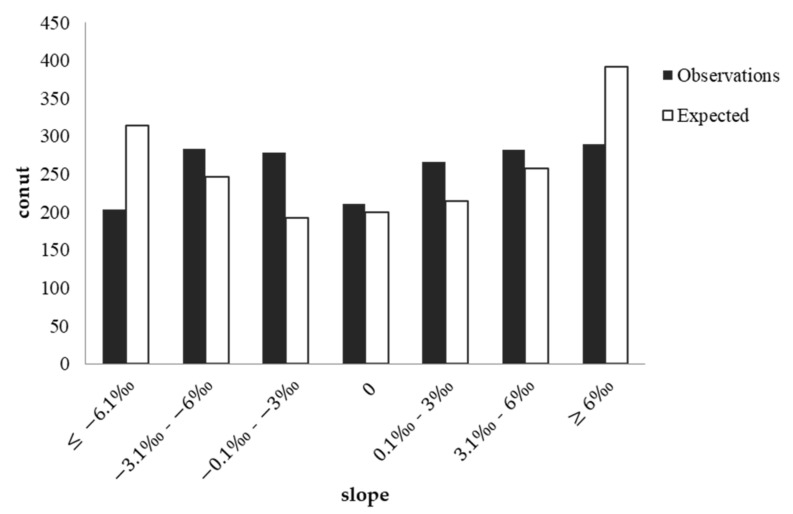
Plot of observed and expected collision occurrence at different slopes.

**Figure 4 ijerph-19-06104-f004:**
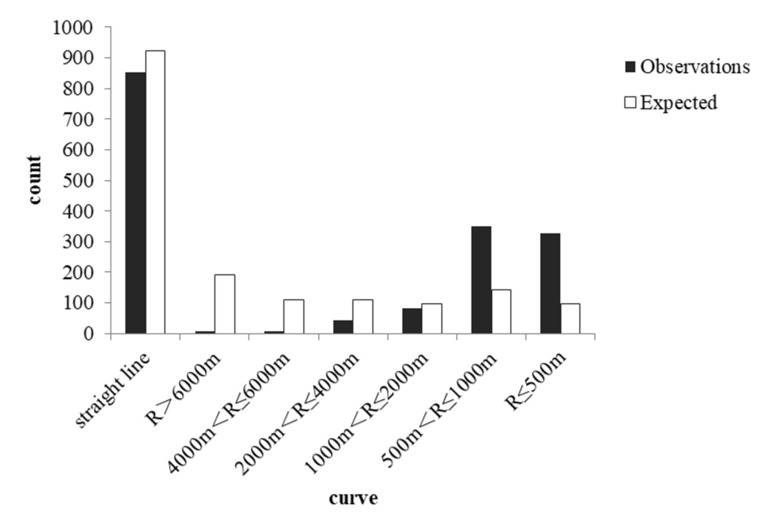
Plot of observed and expected accident occurrence with different curve radii.

**Figure 5 ijerph-19-06104-f005:**
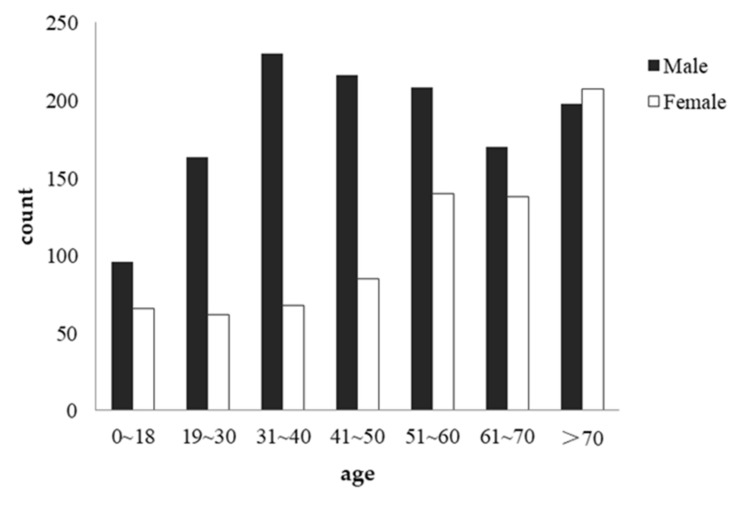
Collision frequency across different sex and age groups.

**Figure 6 ijerph-19-06104-f006:**
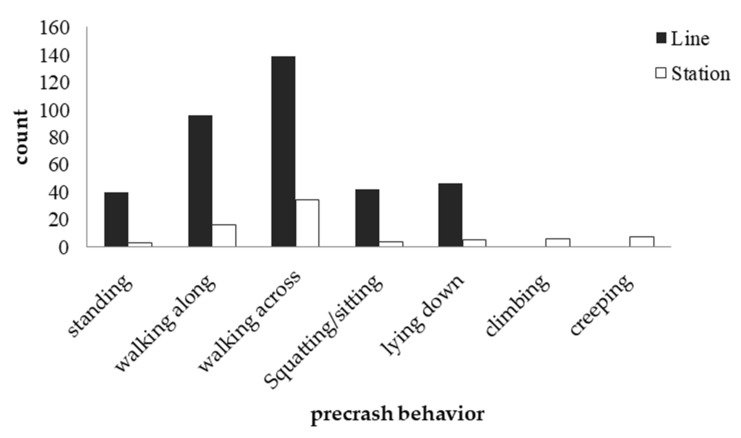
Collision frequency across different behaviors and locations.

**Figure 7 ijerph-19-06104-f007:**
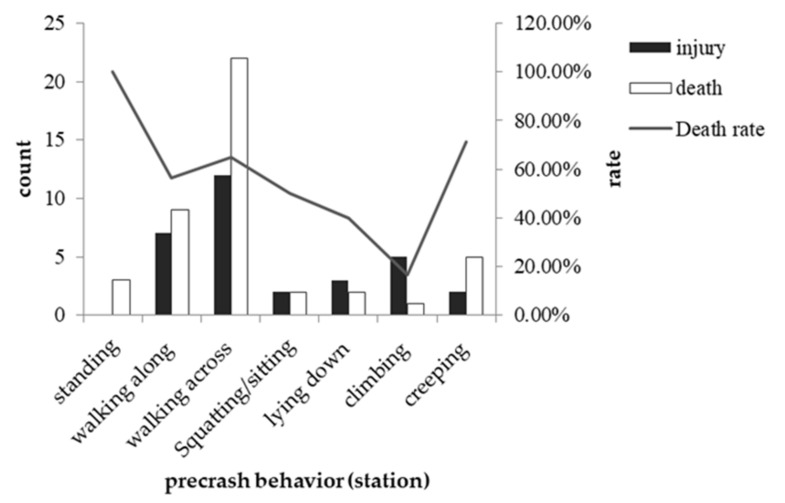
Occurrence of different crash severities by different precrash behaviors in the station.

**Figure 8 ijerph-19-06104-f008:**
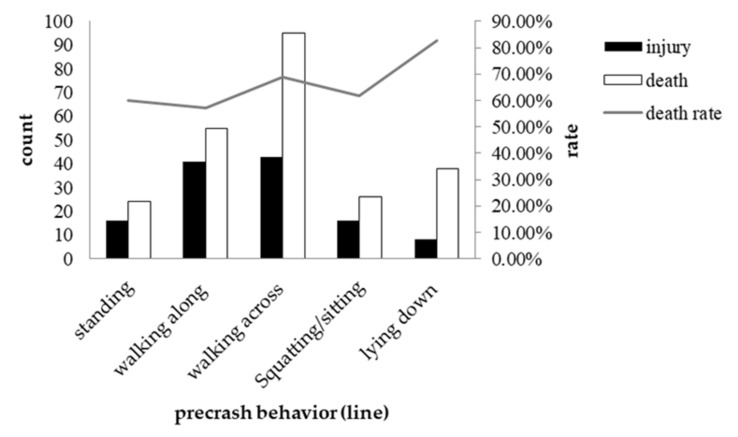
Occurrence of different crash severities by different precrash behaviors along the line.

**Figure 9 ijerph-19-06104-f009:**
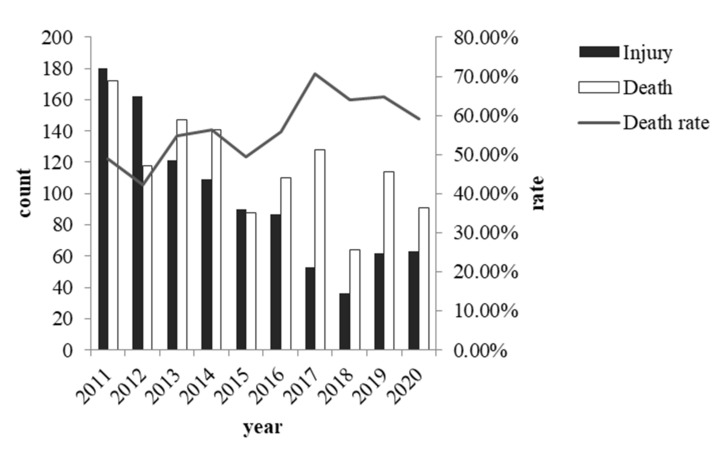
Occurrence of different collision severities in different years.

**Figure 10 ijerph-19-06104-f010:**
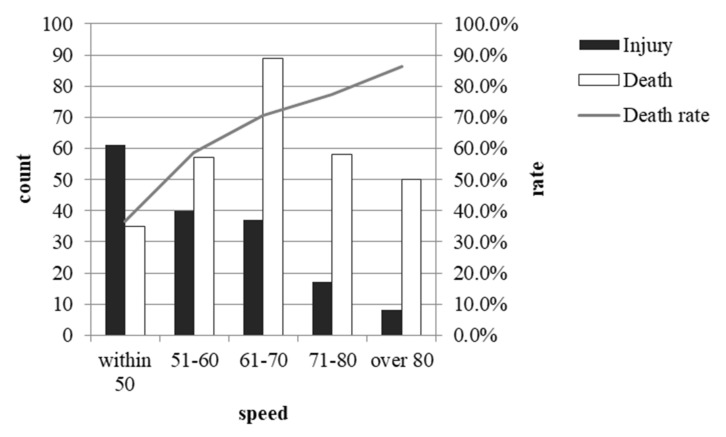
Frequency of different collision severities with different speeds.

**Table 1 ijerph-19-06104-t001:** Partial summary of studies on influencing factors.

Factors	Description	Source
**Time**		
Month	Train–pedestrian fatalities were evenly distributed by month.	Silla and Luoma, 2012 [18]
	27% of unintentional crossing and station deaths occurred in June. Suicide peak is in the spring.	Savage, 2016 [24]
	December has the most trespasser injuries of all months, and June has the least.	Patterson, 2004 [25]
Day of week	The majority of trespasser injuries occurred between Wednesday and Saturday.	Patterson, 2004 [25]
	Railroad suicides mainly occurred on Thursday and Friday.	Krysinska and De Leo, 2008 [20]
Time of day	Railway suicides often happened in the afternoon and evening and after midnight.	Silla and Luoma, 2012 [18]
	The time that trespassers are killed or injured is reasonably evenly spread across the day.	Patterson, 2004 [25]
**Pedestrians**		
Sex	Most victims were male in all types of fatal train–pedestrian collisions.	Savage, 2007; Silla and Luoma, 2012; Rådbo et al., 2005; Erazo et al., 2004 [15,16,17,18]
Age	People between the ages of 16 and 45 faced the greatest risk.	Savage, 2007 [17]
	The majority were 21–40 years old.	Mohanty, et al., 2007 [19]
	51.4% of all collisions happened to people aged 10–29 years.	Silla and Luoma, 2012 [18]
Precrash behaviors	Most precrash behavior was walking, sitting or lying on the track.	Office of Railroad Safety, 2013 [21]
Alcohol or drug related	Many train–pedestrian collisions involved alcohol or drugs.	Centers for Disease Control and Prevention, 1999; Silla and Luoma, 2012; Carias, et al., 2020 [18,22,23]
Place of residence	Few victims were homeless or transients.	Pelletier, 1997 [5]
**Place**		
Fence	After creating awareness by interventions and repairing fences, a substantial decrease in the trespassing rate was observed.	Lobb et al., 2006 [3]
Level crossing	The density of grade crossings and stations increased the frequency of unintentional deaths.	Savage, 2016 [24]
	The correlates of injury severity differ across highway-rail grade crossings and non-crossings.	Zhang et al., 2018 [28]
**Train**		
Train type	More than 80% of fatalities occurred on routes where there was a commuter rail service, and 60% of fatalities involved commuter trains.	Savage, 2016 [24]

**Table 3 ijerph-19-06104-t003:** Classification of precrash behaviors.

Precrash Behaviors Categories (Abbreviation)
Standing on the shoulder or between the tracks (standing)
2. Walking along the track (walking along)
3. Walking or running across the track (walking across)
4. Squatting or sitting on the track or between the tracks (squatting/sitting)
5. Lying down on or between the tracks (lying down)
6. Climbing over the train (climbing)
7. Creeping through the bottom of the train (creeping)

**Table 4 ijerph-19-06104-t004:** Statistical results of all single variables.

*Variables*	*Count*	*Proportion*	*Variables*	*Count*	*Proportion*	*Variables*	*Count*	*Proportion*	*Variables*	*Count*	*Proportion*
** *Severity* **			** *Month* **			* **Time of day** *			** *Precrash behavior* **		
*injury*	963	45.10%	*January*	178	8.30%	0:00–0:59	50	2.30%	*standing on the shoulder/between the tracks*	43	9.80%
*death*	1173	54.90%	*February*	132	6.20%	1:00–1:59	44	2.10%	*walking along the track*	112	25.60%
** *Age* **			*March*	166	7.80%	2:00–2:59	53	2.50%	*walking/running across the track*	173	39.50%
*0–18*	167	8.10%	*April*	176	8.20%	3:00–3:59	37	1.70%	*squatting/sitting on/between the tracks*	46	10.50%
*18.1–30*	226	11.00%	*May*	180	8.40%	4:00–4:59	38	1.80%	*lying down on/between the tracks*	51	11.60%
*30.1–40*	298	14.50%	*June*	172	8.00%	5:00–5:59	54	2.50%	*climbing over the train (in station only)*	6	1.40%
*40.1–50*	301	14.70%	*July*	197	9.20%	6:00–6:59	69	3.20%	*creeping under the train (in station only)*	7	1.60%
*50.1–60*	348	16.90%	*August*	192	9.00%	7:00–7:59	103	4.80%	** *Train type* **		
*60.1–70*	309	15.00%	*September*	169	7.90%	8:00–8:59	113	5.30%	*operating trains*	46	2.20%
*>70*	405	19.70%	*October*	182	8.50%	9:00–9:59	91	4.30%	*high-speed passenger trains*	11	0.50%
** *Sex* **			*November*	181	8.50%	10:00–10:59	141	6.60%	*regular-speed passenger trains*	786	37.10%
*male*	1334	62.90%	*December*	214	10.00%	11:00–11:59	107	5.00%	*freight trains*	1275	60.20%
*female*	788	37.10%	** *Workday* **			12:00–12:59	121	5.70%	* **Curve radius (R)** *		
** *Fence* **			*Monday*	334	15.60%	13:00–13:59	106	5.00%	*straight line*	852	51.10%
*yes*	77	10.00%	*Tuesday*	278	13.00%	14:00–14:59	130	6.10%	*R* > 6000 m	6	0.40%
*no*	694	90.00%	*Wednesday*	308	14.40%	15:00–15:59	118	5.50%	4000 m < *R* ≤ 6000 m	8	0.50%
** *Year* **			*Thursday*	301	14.10%	16:00–16:59	118	5.50%	2000 m < *R* ≤ 4000 m	44	2.60%
*2011*	352	16.50%	*Friday*	300	14.00%	17:00–17:59	115	5.40%	1000 m < *R* ≤ 2000 m	81	4.90%
*2012*	282	13.20%	*Saturday*	295	13.80%	18:00–18:59	105	4.90%	500 m < *R* ≤ 1000 m	349	20.90%
*2013*	268	12.50%	*Sunday*	322	15.10%	19:00–19:59	110	5.10%	*R* ≤ 500 m	327	19.60%
*2014*	250	11.70%	** *Speed* **			20:00–20:59	93	4.30%	* **Slope** *		
*2015*	178	8.30%	*within 50*	96	21.20%	21:00–21:59	81	3.80%	≤−6.1‰	204	11.20%
*2016*	198	9.30%	*51–60*	97	21.40%	22:00–22:59	66	3.10%	−3.1‰–−6‰	284	15.60%
*2017*	181	8.50%	*61–70*	127	28.00%	23:00–23:59	76	3.60%	−0.1‰–−3‰	279	15.40%
*2018*	100	4.70%	*71–80*	75	16.60%	** *Location* **			*0*	211	11.60%
*2019*	176	8.20%	*>80*	58	12.80%	*station*	231	10.80%	0.1‰–3‰	266	14.70%
*2020*	154	7.20%	** *Level crossing* **			*line*	1908	89.20%	3.1‰–6‰	282	15.50%
			yes	2	0.09%				≥6‰	289	15.90%
			no	2137	99.91%						

**Table 5 ijerph-19-06104-t005:** Distribution of different slope ranges in the total mileage and collisions.

Slope Categories	Mileage (km)	Percentage of Total Mileage	Percentage of Collisions
slope < −6‰	3101.104	17.30%	11.20%
−6‰ ≤ slope < −3‰	2432.005	13.60%	15.60%
−3‰ ≤ slope < 0	1892.944	10.60%	15.40%
slope = 0	1967.334	11.00%	11.60%
0 < slope ≤ 3‰	2107.16	11.80%	14.70%
3‰ < slope ≤ 6‰	2537.157	14.20%	15.50%
slope > 6‰	3856.181	21.60%	15.90%
Total	17,893.88	100%	100%

**Table 6 ijerph-19-06104-t006:** Distribution of different curve radii in the total mileage and collisions.

R Categories	Mileage (km)	Percentage of Total Mileage	Percentage of Collisions
straight line	9916.817	55.40%	51.10%
R > 6000 m	2035.567	11.40%	0.40%
4000 m < R ≤ 6000 m	1180.776	6.60%	0.50%
2000 m < R ≤ 4000 m	1173.819	6.60%	2.60%
1000 m < R ≤ 2000 m	1025.104	5.70%	4.90%
500 m < R ≤ 1000 m	1543.594	8.60%	20.90%
R ≤ 500 m	1018.207	5.70%	19.60%
Total	17,893.88	100%	100%

**Table 7 ijerph-19-06104-t007:** Behavior statistics at stations and along lines.

Precrash Behavior Category	Station		Line	
	Count	% of Behavior	Count	% of Behavior
Standing	3	4.00%	40	11.02%
2. Walking along	16	21.33%	96	26.45%
3. Walking across	34	45.33%	139	38.29%
4. Squatting/sitting	4	5.33%	42	11.57%
5. Lying down	5	6.67%	46	12.67%
6. Climbing	6	8.00%	0	0.00%
7. Creeping	7	9.33%	0	0.00%

**Table 8 ijerph-19-06104-t008:** Collision fatality rates by location and behavior.

Precrash Behavior Category	Station			Line		
	Injury	Death	% of Deaths	Injury	Death	% of Deaths
Standing	0	3	100.00%	16	24	60.00%
2. Walking along	7	9	56.25%	41	55	57.29%
3. Walking across	12	22	64.71%	43	95	68.84%
4. Squatting/sitting	2	2	50.00%	16	26	61.90%
5. Lying down	3	2	40.00%	8	38	82.61%
6. Climbing	5	1	16.67%	0	0	—
7. Creeping	2	5	71.43%	0	0	—
Total	31	44	58.67%	124	238	65.75%

## Data Availability

Restrictions apply to the availability of these data. Data were obtained from China Railway Chengdu Bureau Group Co., Ltd. and are available from Jingyu Zhang with the permission of China Railway Chengdu Bureau Group Co., Ltd.

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
