# Peer review of "Characteristics of Train–Pedestrian Collisions in Southwest China, 2011–2020"

_ijerph, 2022, doi:10.3390/ijerph19106104_

Round 1
Reviewer 1 Report
This paper explores the characteristics of accidents between trains and pedestrians in southwest China. It makes use of simple descriptive statistics to show accidents data patterns and associations among some variables.
The data analysis protocol should be improved since the process has not been clearly described (e.g. it is stated that "We first analysed the accident rates along with each variable."; however, chapter 3.1 refers to "accident frequency", not accident rate. Also, the "Interaction between variables to influence accident occurrence" is missing).
The available number of records for each variable is not constant; in some cases, information is only available for 21% of all cases (see table 2). Especially for these cases, it should be pointed out if the sub-population has limitations that might affect results, e.g. if the accidents for which the speed is known only refer to accidents on the line.
The manuscript is overall well-written; just a few typos have been identified (e.g. Figure 6 legend).
Detailed comments
Lines 210-211: The injuries distribution by age and gender should be compared to the population distribution of southwest China, not the full country distribution.
Lines 253-255: The sentence is not clear. According to Figure 10, the "upward trend" seems to be related to the death rate, not the number of casualties.
Lines 275:286: The result is interesting. I was wondering if there is a possibility to also check the location of train tracks in relation to the level of urbanisation as this could be a confounding factor. I think that routes in or near urban areas have a higher level of exposure than routes outside urban areas.
Figure 8: This figure is referring to pre-crash behaviour in stations; what about the line?
Figure 9: while the number of deaths and injuries is declining over the period, it seems the death rate is increasing. Is there any explanation for this?
Chapter 3.3: The analysis between location and severity is missing.
Lines 486-495: The accident rate at level crossings is very different from other countries. Is there any measure of exposure that could explain this difference? For example, a measurement of the density of level crossings on the network.
Author Response
Response to Reviewer 1 Comments
Point 1: The data analysis protocol should be improved since the process has not been clearly described (e.g. it is stated that "We first analyzed the accident rates along with each variable."; however, chapter 3.1 refers to "accident frequency", not accident rate. Also, the "Interaction between variables to influence accident occurrence" is missing.
Response 1: Thanks for your suggestion. We have modified the expressions in Chapter “2.3 Data analysis protocol” accordingly.
Point 2: The available number of records for each variable is not constant; in some cases, information is only available for 21% of all cases (see table 2). Especially for these cases, it should be pointed out if the sub-population has limitations that might affect results, e.g. if the accidents for which the speed is known only refer to accidents on the line.
Response 2: Thanks for your suggestion. We delineated the percentage of missing values of each variable in the method part more clearly by including possible reasons behind the failure to collect all data. We also pointed out possible limitations by drawing conclusions from these incomplete data. For example, the reason why speed is missing was due to simplified recording requirements in early years (ca. 2011-2015). Therefore, the available data were more likely to appear in the records of recent years. As a result, we must be aware any conclusions about the speed might be confined to situations of recent years.
Point 3: The manuscript is overall well-written; just a few typos have been identified (e.g. Figure 6 legend).
Response 3: Thanks for your suggestion. We have checked the full manuscript again and corrected the typos.
Point 4: Lines 210-211: The injury distribution by age and gender should be compared to the population distribution of southwest China, not the full country distribution.
Response 4: Thanks for your suggestion. We have replaced the population data for comparison with those in the greater Sichuan-Chongqing area.
Point 5: Lines 253-255: The sentence is not clear. According to Figure 10, the "upward trend" seems to be related to the death rate, not the number of casualties.
Response 5: Sorry for causing this confusion, we misplaced a description on the severity in this paragraph. In the revision, we rewrote this sentence into “The frequency of crashes in different speed zones is different”.
Point 6: Lines 275:286: The result is interesting. I was wondering if there is a possibility to also check the location of train tracks in relation to the level of urbanisation as this could be a confounding factor. I think that routes in or near urban areas have a higher level of exposure than routes outside urban areas.
Response 6: Thanks for your suggestion. Actually, we are running this analysis right now. However, we met an unexpected difficulty. The location of the incidents was not recorded using normal longitude and latitude coordinates, but an internal line-section system. It took a great time to build a transformation software, so we plan to include this analysis in a future paper by also investigating other higher level (e.g., county level) features.
Point 7: Figure 8: This figure is referring to pre-crash behavior in stations; what about the line?
Response 7: Thanks for your suggestion. In this revision, we've added a figure depicting the relationship between precrash behavior along the line and crash severity.
Point 8: Figure 9: while the number of deaths and injuries is declining over the period, it seems the death rate is increasing. Is there any explanation for this?
Response 8: Thanks for your suggestion. We have explained this trend, first of all, with the development of technology, the speed of trains has gradually increased during this decade, which may increase the severity of collisions. However, we cannot prove this assertion using our own data directly because the speed in early years were not fully recorded, as mentioned in a previous response. Secondly, more protective nets were installed after 2014. While this can reduce the number of less motivated trespassers (e.g. who only want to take a shortcut), it might not reduce the number of very motivated intruders. For example, some people may deliberately break the protective net to commit suicide or just showing off. As these behaviors are more related to severe consequences, the increase of death rate is reasonable. A minor but also possible reason is driver may reduce their degree of vigilance when driving along the line with protective nets installed. They would not expect people to appear on the track and have slower reactions. In the revision, we added such an explanation in the discussion part.
Point 9: Chapter 3.3: The analysis between location and severity is missing.
Response 9: Thanks for your suggestion. The analysis results between location and severity have been added to 3.3. We divided the location variable into the track feature category. According to our results, location has no significant effect on the severity of the accident.
Point 10: Lines 486-495: The accident rate at level crossings is very different from other countries. Is there any measure of exposure that could explain this difference? For example, a measurement of the density of level crossings on the network.
Response 10: Thanks for your suggestion. From the documents we obtained from the China Railway Chengdu Group Co., Ltd, we found there are 48 level crossings along all the tracks in this area (17,893.88 kilometers). Therefore, the number of level crossings per 100 kilometers is 0.268. According to public data in the United States, the United States has a total of 203,778 level crossings (about 250,000 kilometers), and the number of level crossings per 100 kilometers is 81.5, which is much higher than China. In this revision, we provided this information and made a short discussion in the discussion part.

Reviewer 2 Report
This work is straightforward - a series of univariate hypothesis tests on the various factors in train-pedestrian collisions. The authors' approach assumes these factors in train-pedestrian collisions are independent. This is the primary weakness of the univariate approach taken by the authors - these factors in train-pedestrian collisions may actually be related, and their covariances should be considered in the hypothesis testing. Thus, this work would benefit greatly from multivariate hypothesis testing to account for the covariances between these factors.
Author Response
Response to Reviewer 2 Comments
Point 1: This work is straightforward - a series of univariate hypothesis tests on the various factors in train-pedestrian collisions. The authors' approach assumes these factors in train-pedestrian collisions are independent. This is the primary weakness of the univariate approach taken by the authors - these factors in train-pedestrian collisions may actually be related, and their covariances should be considered in the hypothesis testing. Thus, this work would benefit greatly from multivariate hypothesis testing to account for the covariances between these factors.
Response 1: Thanks for pointing out this weakness. We fully agree with the reviewers that future studies may benefit from using multivariate analysis to dig deeper into this data set. Indeed, we used some types of multivariate analysis. For example, we examined the joint influence of gender and age. However, for two reasons, we think we must refrain from using more sophisticated multivariate techniques in this present paper. First, different from previous studies conducted in the western world, the present study provides many new perspectives, some were due to new types of data available (e.g. slopes, curvature), and some were due to different cultural and social contexts. We think this information can provide many insights by just presenting them using simple descriptive statistics. Such an approach can provide a full picture of the phenomenon and is more compatible with the exploratory nature of this paper. Many new hypotheses derived from this study can be explored with a more refined methodology in the future. Second, as not all types of data were available, using any form of multivariate analysis may suffer from the difficulty to establish a robust joint distribution. If we have to use multivariate analysis, it means we have to exclude a certain amount of data, which is not in accordance with the main purpose of this paper.

Reviewer 3 Report
I commend the authors for this original research that investigates the train-pedestrian collisions—an important safety topic. To improve the manuscript, I have a few suggestions as follows:
Introduction:
Table 1: The latest study reviewed here is from year 2016. Have there been any more recent studies on the topic?
Lines 64, 70, 113, 171, and throughout the manuscript: Please use “collision” or “crash” instead of “accident” and please be consistent with whichever word you choose. Currently, there are three different words (collision, accident, and crash) used in various places within the manuscript referring to the same incident (the incident of train-pedestrian collisions).
Methods:
As a general comment on the methodology section: I think your study would benefit from a regression model such as a Poisson model that can examine the relationship between the factors you listed in Table 2 and the number/severity level of train-pedestrian collisions. That way, we would know more about the contributing factors that play a role in such collisions and the significant correlations that may exist. Any particular reason why you chose to only conduct a descriptive statistical analysis and not proceed to explore a statistical model?
(After getting to the “Limitations and future direction” section, I saw that you mentioned the possibility of using a Poisson model to analyze these data in the future. I agree with that idea and encourage you to do so);
Figure 1: I think this figure needs to be larger and more clear. Please add some text to the figure to identify the study area. I know the highlighted region shows the study area on the map; but, the map would be more professional and informative with addition of some text. Also, what are the light blue dots and hashed curves under the map representing?
Table 2: On the line immediately preceding Table 2 (Line 186), you say that you have “four categories”. In the table title, you mention “Variable information divided into five categories.” Please correct the table title to indicate that you have four categories;
Specific comments:
There are a few grammatical/writing issues that can be revised. I provide a few examples below, which in my opinion, can improve the manuscript:
Line 55: Please revise the sentence to “In the United States, the casualty ratio was is 54.1% in 2021 [8];
Line 57: Please choose another word instead of “bloody”;
Lines 60-61: In the sentence “…it is necessary to identify the causes so that administrative personnel can take appropriate…” , please mention the causes of what (please specify whether you meant the causes of pedestrian intrusion or causes of train-pedestrian collisions in this particular sentence);
Lines 325-326: Please provide a reference for the sentence “Statistics show that when a train-pedestrian collision occurs on a railway, it is more likely to cause a death than an injury.”
Author Response
Response to Reviewer 3 Comments
Point 1: Table 1: The latest study reviewed here is from year 2016. Have there been any more recent studies on the topic?
Response 1: Thanks for your suggestion. We have to point out that conducting such studies is difficult across the world. First, not all countries may have such recordings. Second, researchers may have difficulties in getting access to these recordings. Third, the data set must be large enough to make some valuable analysis. In this way, it is not very surprising that similar studies are very few, not comparable to research on road safety. Nevertheless, we have found and added some newer literature to the table (Zhang et al., 2018; Carias, et al., 2020 ).
Point 2: Lines 64, 70, 113, 171, and throughout the manuscript: Please use “collision” or “crash” instead of “accident” and please be consistent with whichever word you choose. Currently, there are three different words (collision, accident, and crash) used in various places within the manuscript referring to the same incident (the incident of train-pedestrian collisions).
Response 2: Thanks for your suggestion. We have distinguished “accident”, “crash” and “collision” in the manuscript and made some corrections.
Point 3: As a general comment on the methodology section: I think your study would benefit from a regression model such as a Poisson model that can examine the relationship between the factors you listed in Table 2 and the number/severity level of train-pedestrian collisions. That way, we would know more about the contributing factors that play a role in such collisions and the significant correlations that may exist. Any particular reason why you chose to only conduct a descriptive statistical analysis and not proceed to explore a statistical model?(After getting to the “Limitations and future direction” section, I saw that you mentioned the possibility of using a Poisson model to analyze these data in the future. I agree with that idea and encourage you to do so);
Response 3: Thanks for your suggestion. We argue that one important merit of this paper is to show the original form of this phenomenon (the train-pedestrian collision in China) without losing or distorting any information. In addition, we provided many new types of information and insights to strengthen the exploratory nature of this paper. We do plan to use regression analysis in the future. But due to data missing, we cannot include all related variables in the analysis. So we have to sacrifice certain amount of information in answering some more specific questions. While these studies are also valuable (and we are doing right now), we think they are different from the current paper in many ways. We can also expand the present paper to include these analyses, but it would make the paper very lengthy and redundant. As a result, we would be more willing to use this approach to address more specific and focused questions in the future, possibly a separate paper.
Point 4: Figure 1: I think this figure needs to be larger and more clear. Please add some text to the figure to identify the study area. I know the highlighted region shows the study area on the map; but, the map would be more professional and informative with addition of some text. Also, what are the light blue dots and hashed curves under the map representing?
Response 4: Thanks for your suggestion. Figure 1 has been adjusted.
Point 5:Table 2: On the line immediately preceding Table 2 (Line 186), you say that you have “four categories”. In the table title, you mention “Variable information divided into five categories.” Please correct the table title to indicate that you have four categories;
Response 5: Thanks for your suggestion. We have changed “five” to “four” in the title of Table 2.
Point 6: Line 55: Please revise the sentence to “In the United States, the casualty ratio was is 54.1% in 2021 [8];
Response 6: Thanks for your suggestion.We have revised this sentence as suggested.
Point 7: Line 57: Please choose another word instead of “bloody”;
Response 7: Thanks for your suggestion. We have replaced the word “bloody” with “dreadful”.
Point 8: Lines 60-61: In the sentence “…it is necessary to identify the causes so that administrative personnel can take appropriate…”, please mention the causes of what (please specify whether you meant the causes of pedestrian intrusion or causes of train-pedestrian collisions in this particular sentence);
Response 8: Thanks for your suggestion. We elaborate on this sentence in more detail, noting that the cause refers to the cause of the train-pedestrian collision.
Point 9: Lines 325-326: Please provide a reference for the sentence “Statistics show that when a train-pedestrian collision occurs on a railway, it is more likely to cause a death than an injury.”
Response 9: Thanks for your suggestion. This was an improper use of English. Our original intention was to express that from our data, we can see that in train-pedestrian collisions, the probability of death is higher than the probability of injury. So we changed this expression to “our results showed that .......”

Reviewer 4 Report
This manuscript presents an interesting study to analyze the characteristics associated with train-pedestrian collisions in southwest China. For this purpose, a descriptive statistical analysis was performed on a larger dataset of traffic collisions between 2011 and 2020.
More advanced techniques (like data mining) could have been included to obtain more specific results.
Minor comment: Figures 2 and 3 could be merged into a single figure (one on the right and one on the left).
Author Response
Response to Reviewer 4 Comments
Point 1: More advanced techniques (like data mining) could have been included to obtain more specific results.
Response 1: Thanks for your suggestion. The core of this research is to use descriptive statistics to display the original distribution of the phenomenon and provide insights and new hypotheses for future studies to explore further. Future studies may benefit from using multivariate analysis to dig deeper into this data set. However, for two reasons, we think we must refrain from using more sophisticated multivariate techniques in this present paper. First, different from previous studies conducted in the western world, the present study provides many new perspectives, some were due to new types of data available (e.g. slopes, curvature), and some were due to different cultural and social contexts. We think this information can provide many insights by just presenting them using simple descriptive statistics. Such an approach can provide a full picture of the phenomenon and is more compatible with the exploratory nature of this paper. Many new hypotheses derived from this study can be explored with a more refined methodology in the future. Second, as not all types of data were available, using any form of multivariate analysis may suffer from the difficulty to establish a robust joint distribution. If we have to use multivariate analysis, it means we have to exclude a certain amount of data, which is not in accordance with the main purpose of this paper.
Point 2: Minor comment: Figures 2 and 3 could be merged into a single figure (one on the right and one on the left).
Response 2: Thanks for your suggestion. We have merged Figure 2 and 3.
